# Simulation Study on the Influence of Different Molten Steel Temperatures on Inclusion Distribution under Dual-Channel Induction-Heating Conditions

**DOI:** 10.3390/ma16247556

**Published:** 2023-12-08

**Authors:** Bing Yi, Guifang Zhang, Qi Jiang, Peng Yan, Zhenhua Feng, Nan Tian

**Affiliations:** 1Faculty of Metallurgical and Energy Engineering, Kunming University of Science and Technology, Kunming 650093, China; yibing8578@163.com (B.Y.); yanp_km@163.com (P.Y.); fengzhenhua666@126.com (Z.F.); tian1852558@163.com (N.T.); 2Hunan Zhongke Electric Co., Ltd., Yueyang 414000, China

**Keywords:** induction heating, inclusion particles, temperatures, mathematical simulation

## Abstract

Impurity elimination in tundishes is an essential metallurgical function in continuous casting. If inclusions in a tundish cannot be effectively removed, their presence will have a serious impact on the quality of the bloom. As a result, this research investigates the locations of inclusion particles in a six-strand induction-heating tundish in depth, combining the flow, temperature, and inclusion trajectories of molten steel under electromagnetic fields. The results show that a pinch effect occurred in the induction-heating tundish, and a rotating magnetic field formed in the channel, with a maximum value of 0.158 T. The electromagnetic force was directed toward the center of the axis, and its numerical distribution corresponds to the magnetic flux density distribution, with a maximum value of 2.11 × 10^5^ N/m^3^. The inclusion particles’ movement speed accelerated as the molten steel’s temperature rose, and their distribution in the channel was identical to the rotating flow field distribution. When the steel’s temperature rose from 1750 K to 1850 K, the removal percentage of inclusion particles in the discharge chamber rose by 9.20%, the removal rate at the outlet decreased from 8.00% to 3.00%, and the adhesion percentage of inclusion particles in the channel decreased from 48.40% to 44.40%.

## 1. Introduction

Tundish metallurgy, a crucial link in the steelmaking process, entails the performance of vital metallurgical functions such as enhancing molten steel cleanliness and controlling molten steel temperature [1,2]. Sometimes, the finished steel has issues such as poorly controllable quantity and problems regarding the particle locations of nonmetallic inclusions, causing the inclusions to be inherited by the cast bloom, negatively impacting the chemical and mechanical characteristics of the resulting steel [3,4,5].

Many scholars have conducted research and reports on the elimination of nonmetallic inclusions in tundishes. For example, Solorio et al. [6] used computational and physical simulations to investigate the differences in the molten steel flow and inclusion loss rate in a tundish using a traditional ladle shroud and a swirling ladle shroud. Furthermore, the swirling ladle shroud, an improved flow control shroud, could reduce the formation of vortices and the backflow of molten steel under a uniform and uneven temperature distribution, as well as enhance the rotation speed of inclusions, which is helpful for inclusion removal. Zhang et al. [7] investigated the transmission of heat and flow, inclusion elimination in molten steel in a four-strand tundish, and the effect of strand obstruction on the transmission of heat and flow. A converter design that permitted inclusion removal was also presented. Moreover, they also investigated the impact of alternative flow control retaining wall designs in the tundish on the flow field and inclusion removal [8]. The results revealed that the flow control retaining wall setting is conducive to inclusion elimination and that the generation of collisions with inclusions and their adherence to the furnace lining are the major approaches effective for removing smaller inclusions. Similarly, Wang et al. [9] investigated how multi-hole ceramic filters impact molten steel flow and inclusion particle trajectory in tundishes. According to the findings reported by Xing et al. [10,11], increasing the power of the induction heating equipment benefits the temperature increase in the receiving chamber while also progressively increasing the elimination rate of inclusion particles in the receiving chamber. Furthermore, as compared to a straight-channel tundish, the inclusion removal rate in the channel region of the novel curved-channel induction-heating tundish was greatly enhanced. According to Ding et al.’s simulation investigation [12], casting speed has little association with the volume fraction of a tundish’s blocked, dead, and entirely mixed zones. Furthermore, there is a linear relationship between the inclusion elimination rate, the peak time of the response curve, and the steel’s minimum residency duration in the tundish.

In summary, the current research on inclusion removal in tundishes is primarily concerned with the elimination of inclusions in single-strand or double-strand tundishes, the impact of flow-control mechanisms such as retaining walls and multi-hole ceramic flow restrictors on inclusion removal, and the impact of smelting conditions such as casting speed. However, there are few findings on the elimination of inclusions in dual-channel induction-heating tundishes. Recently, Wang et al. [13,14] carried out some research in this field, concentrating on the effect of an induction-heating tundish’s channel diameter on the inclusion clearance rate. In real conditions, the molten steel temperature will fluctuate depending on the steel type and casting speed [15,16]. Furthermore, there is currently no existing research on the impact of molten steel temperature difference on the distribution and deletion of inclusion particles within a tundish.

Based on the above, this paper establishes a three-dimensional geometric model of a dual-channel induction-heating tundish, combining the magnetic field, flow field, temperature field, and particle-tracking physical fields. The magnetic field, flow field, and temperature distribution in the tundish under electromagnetic induction-heating conditions and the influence of different molten steel temperatures on the distribution of inclusion particles in the double-channel induction-heating tundish was studied. 

Based on above, the flow field, temperature field, magnetic field, and particle-tracking physical fields are all combined in this paper’s three-dimensional geometric model of a double-channel induction-heating tundish. A detailed investigation of magnetic flux density, flow trend, molten steel temperature variations, and the effect of molten steel temperature on inclusion trajectory is provided, offering a theoretical reference for increasing molten steel cleanliness in a double-channel induction-heating tundish.

## 2. Mathematical Model Description

### 2.1. Geometry and Meshing

#### 2.1.1. Geometry

The tundish geometry used in this study is shown in Figure 1. This tundish is suitable for casting blooms with section dimensions of 380 × 280 mm^2^, and the corresponding casting speed is 0.68 m/min. The tundish was split into a flow-receiving chamber, channel zone, and discharging chamber based on its diverse roles and forms in order to enhance the heating and stirring effects of molten steel, as well as allow for the proper positioning of induction-heating equipment, as shown in the description in Figure 1a,b. The location of each outlet was marked in Figure 1b. Table 1 shows the precise geometric dimensions of the induction-heating tundish. Table 1 and Figure 1c show that the channel has the shape of a double fork. Both the main channel and the sub-channel have the same radius of 80 mm. The main channel and sub-channel were raised by 3° and 15°, respectively. The upward-angled channel design facilitates complete molten steel flow and eliminates impurities in the discharging chamber.

#### 2.1.2. Meshing

Tetrahedral mesh was used to divide the tundish’s three-dimensional geometric mesh because it can adapt well to various shapes. To enhance the calculation results for the channel, it was separated into regular cylinders and irregular sections of the bifurcation by using the extension surface of the discharging chamber wall as a cross-section. The regular cylindrical part was swept with a triangular mesh, and the irregular part was divided into a tetrahedral mesh. At the same time, the boundary layer meshing of the tundish wall was performed. On the basis of the finer grid division of the core and coil, the focus here was on grid-independent research on molten steel. The number of tundish meshes was divided into 800,000; 1,200,000; 1,600,000; and 2,000,000. According to the computation findings, the average steel flow rate in the tundish is 0.021 m/s when there are 1,600,000 elements, and the average flow rate of molten steel in the tundish stays constant if the mesh is improved further. As a result, the ideal grid number for the tundish in this study should be around 1,600,000. This results in a total of 1,595,860 elements for meshing the molten steel in the induction-heating tundish. Figure 2 depicts a schematic representation of the geometric meshing.

### 2.2. Assumptions

The metallurgical process taking place in a tundish during on-site casting is very complex and cannot be observed. During the study process, the following model assumptions were developed in an effort to increase the convergence of the model and the resemblance between the mathematical model and the real metallurgical process:Molten steel is an incompressible Newtonian fluid.The effect of slag on molten steel is not considered.The molten steel flow is steady.The collision growth of inclusion particles is not taken into account.The inclusion particles are eliminated when they reach the top layer of the molten steel.

### 2.3. Governing Equations

The Maxwell equation is used to compute the electromagnetic field. The molten steel in the tundish flows according to the continuity equation, the Navier–Stokes equation, and the *k*-*ε* equation, which are applied to the calculation of flow field physics. The energy conservation equation governs the heat transmission of molten steel [17,18] and is used in fluid heat transfer physics. 

The motion state of inclusion particles is determined according to the “particle tracing in fluid flow” physical field, and the relevant formulas used in the physical field are as follows. The momentum equation of inclusion particles in the tundish is expressed in Equation (1) [19]:(1)$$\rho_{P}\frac{\pi }{6}d_{P}^{3}\frac{dv_{P}}{dt}=F_{g}+F_{b}+F_{d}+F_{p}+F_{t}+F_{s}$$

Gravity ($F_{g}$), buoyancy force ($F_{b}$), drag force ($F_{d}$), electromagnetic pressure force ($F_{p}$), thermophoretic force ($F_{t}$), and Saffman force ($F_{s}$) all act on the inclusion particles in the tundish under induction heating.

Since the inclusion particles are non-metallic, without conductivity, the electromagnetic pressure force can be expressed by Equation (2) [20,21]
(2)$$F_{p}=-\frac{\pi d_{p}^{3}}{8}F$$
where $d_{p}$ represents inclusion particles’ diameter, given in m. 

Stokes’ drag force rule applies because the inclusion particles are extremely tiny and the flow speed of the inclusion particles relative to the molten steel is very minimal. Equations (3) and (4) present an expression for determining the drag force [4,22]
(3)$$F_{d}=\frac{u-u_{p}}{\tau_{r}}$$
(4)$$\tau_{r}=\frac{\rho_{p}d_{p}^{2}}{18\mu }$$
where u denotes molten steel’s flow velocity, given in m/s; $u_{p}$ denotes the inclusion particles’ flow velocity, given in m/s; $\rho_{p}$ is the inclusion particles’ density, expressed in kg/m^3^; μ is molten steel’s viscosity, given in Pa·s; and the residency time of inclusions is represented by $\tau_{r}$, s.

The total force of buoyancy and gravity acting on inclusion particles is expressed explicitly in Equation (5) [23,24].
(5)$$F_{g}+F_{b}=\frac{\left(\rho -\rho_{p}\right)g}{\rho_{p}}$$

Here, ρ indicates molten steel’s density in kg/m^3^; g is gravitational acceleration, given in m/s^2^.

The thermophoretic force and Saffman force can be defined using Equations (6)–(9)
(6)$$F_{t}=-\frac{\alpha \beta }{M}\nabla T$$
(7)$$M={\left(6\pi \mu d_{p}\right)}^{-1}$$
(8)$$F_{s}=6.46r_{p}^{2}L_{v}\sqrt{\mu \rho \frac{\left|u-u_{p}\right|}{\left|L_{v}\right|}}$$
(9)$$L_{v}=(u-u_{p})\times \left[\nabla \times (u-u_{p})\right]$$
where the radius of inclusion particles is denoted by $r_{p}$ in Equation (8), given in m.

### 2.4. Model-Solving Process and Boundary Conditions

COMSOL Multiphysics 6.1 was utilized for the calculations in this investigation. A frequency-domain-steady-state solver was used to determine the velocity and heat distribution of molten steel under induction heating, and the motion trajectories of inclusion particles at various moments were subsequently computed using a transient solver. The turbulence dispersion model used in the inclusion particle trajectory-tracking process was a continuous random walk, and the simulation process considered virtual mass force. 

According to previous studies and actual on-site conditions, it is known that when molten steel impacts the turbulence controller, most of the inclusions are removed in the discharging chamber [25]. However, in this study, the entrance cross-sections of the two channels were set as the entrance cross-sections of the inclusion particles so as to perform a comprehensive and extensive study of the movement of inclusion particles in the channel and discharging chamber. The number of inclusion particles released was 500. The density of the inclusion particles was 3900 kg/m^3^, and their diameter was 50 μm. The relationship between the inclusion particles and the inner wall of the tundish corresponds to a rebound with a probability of 0.5, which implies that when the probability of this event exceeds 0.5, the inclusions remain on the tundish’s inner wall. In addition to being set as the tundish outlet, the inclusion particle exit also has the discharging chamber’s top layer. When the inclusion particles contact the slag–steel interface, they are removed, constituting the same phenomenon as the 5th assumption. Molten steel has an inlet velocity of 0.68 m/min, a density of 8523–0.8358 T kg/m^3^ [26], a thermal conductivity of 23.5 W/m·K, a heat capacity at constant pressure of 750 J/kg·K, and a dynamic viscosity of 0.19252 exp (49,618/RT) mPa·s [27]. Turbulent kinetic energy ($k_{i}$ = 0.01$u_{i}^{2}$) and dissipation rate ($\varepsilon_{i}=k_{i}^{1.5}/r_{i}$) were estimated through semi-empirical relationships. In the previous expressions, $u_{i}$ represents the flow rate of the water inlet in m/s, and $r_{i}$ represents the inner diameter of the water inlet in m. Then, the heat losses on the upper surface, bottom, side wall, and channel surface of the tundish are 15,000 W/m^2^, 1800 W/m^2^, 4600 W/m^2^, and 2000 W/m^2^, respectively. An iron core and water-cooled copper coils wound with one-way rotation made up the induction-heating apparatus. The relative magnetic permeability of the iron core is 1000. The working frequency of the coil is 50 Hz, and the output is 800 kW. 

## 3. Results and Discussion

### 3.1. Model Validation

#### 3.1.1. Physical Model Verification

Using the similarity principle as a basis, a water simulation experiment was conducted on the six-strand induction-heating tundish. The water model tundish size to prototype tundish size ratio was 1:2, and the Fr numbers in the corresponding locations were equivalent. The convection of molten steel in the tundish is caused by the force acting on the molten steel. The buoyancy force generated via the density difference (temperature difference) causes natural convection, while the inertial force causes forced convection. Therefore, the convection characteristics of molten steel in the tundish are determined by the ratio of buoyancy force ($F_{b}$) to inertia force ($F_{i}$); that is, the tundish accuracy is expressed as follows: (10)$$Zb=\frac{F_{b}}{F_{i}}=\frac{Gr}{{Re}^{2}}=\frac{\beta gL∆T}{v^{2}}$$
where Gr is the Grashof number; Re denotes Reynolds number; and v is the average flow rate of molten steel in the tundish, given in m/s. The temperature difference between the water model inlet and outlet and the actual temperature difference of the molten steel inflow into the tundish were calculated using Equations (11) and (12)
(11)$${\left(\left.\frac{\beta gL∆T}{v^{2}}\right)\right.}_{p}={\left(\left.\frac{\beta gL∆T}{v^{2}}\right)\right.}_{m}$$
(12)$$\frac{{∆T}_{m}}{{∆T}_{p}}=\frac{\beta_{p}}{\beta_{m}}$$
where m denotes model parameter; p represents prototype parameter; β denotes the thermal expansion coefficient, 1/K; L is characteristic size in m; and ∆T means temperature difference, K. 

During the water simulation experiment, the temperature of the water was about 20 °C, and its volume expansion coefficient was 3.67 × 10^−4^ 1/K; the numerical simulation object was molten steel, whose temperature was about 1500 °C and whose volume expansion coefficient was 1.16 × 10^−4^ 1/K. The coefficient of thermal expansion was substituted into Equation (11) to attain the following expression:(13)$${∆T}_{m}=0.316{∆T}_{r}$$

According to experience and the literature, the temperature rise range of the molten steel in the induction-heating tundish channel is 8–13 K. According to Formula (13), the corresponding temperature rise range of the channel in the water model is 2.5–4.1 K. To simulate the effect of channel induction heating, 3 kW heating rods were used in the water model to heat the water in the channel. There was a 2–4 °C difference that formed between the inlet and outlet of the water model channel. And the result satisfies the *Zb* criterion, which is the ratio of buoyancy force ($F_{b}$) to inertia force ($F_{i}$). Figure 3 provides an on-site picture of the water model. 

Since the tundish has a symmetrical structure, the residence time distribution (RTD) curve only analyzes the stimulus response curves of outlets 1–3. Figure 4 depicts the on-site RTD curve as well as the computed RTD curve.

The abscissa in Figure 4 is dimensionless time, which is the ratio of a certain moment in the flow process of molten steel to the theoretical residence time of molten steel in the tundish. And the theoretical residence time of molten steel is the ratio of the volume of molten steel in the tundish to the flow rate of molten steel. The ordinate of Figure 4 is the dimensionless concentration, which is the ratio of the tracer concentration to the total concentration at a certain moment. In Figure 4, the RTD curves of outlet 1 and outlet 3 basically overlap, except for the fact that the maximum dimensionless concentration of outlet 3 is 0.09 higher than that of outlet 1. Since the channel has a double fork, and outlet 2 is located in the area between the two forks, its fluidity is poorer than that of outlet 1 and outlet 3, so the maximum dimensionless concentration of outlet 2 is relatively low. The test findings are fluctuating because of a variety of adjustments and interfering elements in the on-site experimental process. The detailed flow characteristics of molten steel in the tundish are shown in Table 2. 

#### 3.1.2. Mathematical Model Verification

In our earlier work [17], we verified the inclusion particle trajectory-tracking model under electromagnetic induction-heating conditions and demonstrated the model’s dependability from both a qualitative and quantitative perspective, so we will not go into details on this topic here. 

### 3.2. Magnetic Field, Flow, and Heat Distribution of Molten Steel

In the tundish, the melted steel is affected by the Lorentz force since it is comparable to a closed loop conductor under electromagnetic-induction-heating conditions. Due to the Lorentz force induced by the strong axial current, the molten steel will generate a pinch effect [28] in the direction of the radii. The tundish exhibits a revolving flow of molten steel, with the spinning action being more noticeable in the channel area. Figure 5 displays the streamline placement of molten steel due to the symmetry of the tundish. The red curve in the figure indicates the flow trend of molten steel, and the arrow indicates the flow direction.

Figure 5 shows that the molten steel generates rotational flow in the tundish due to the Lorentz force, and the closer one moves to the wall of the electromagnetic induction heating equipment, the more obvious the rotational flow becomes. When the molten steel enters the channel, the pinch effect occurs, and a rotating flow forms along the axis. The molten steel then flows through the bifurcation to generate an opposing flow stream in the discharge chamber. The rotating flow of molten steel is more visible in the middle of the discharge chamber, near to the induction coil location, than in the side sections of the discharging chamber. 

A cross-section of the tundish was created for comprehensive study in order to better explain the distribution of magnetism and temperature in the tundish. A schematic diagram of different sections in the tundish is shown in Figure 6, where section B-B is located in the middle of the core in the XZ direction. Section A-A is the section of the channel’s center, a section with an upward angle of 3°.

As illustrated in Figure 7, the B-B section was used to analyze the magnetic flux density and the distribution of Lorentz force in the channel. Figure 7a shows that the magnetic flux density in the channel turns counterclockwise and is unevenly distributed. On the edge nearest to the coil, the magnetic flux density is higher. The greatest magnetic flux density on the B-B section is 0.158 T, occurring near the coil’s surface, while the minimum value was found on the section’s lower left. Correspondingly, the Lorentz force on the liquid steel is the lowest at the position with the lowest magnetic flux density, as shown in Figure 7b. 

The Lorentz force in the B-B section is directed toward the lower left of the center, and the Lorentz force near the coil surface in the upper right is the largest, amounting to 2.11 × 10^5^ N/m^3^. The Lorentz force progressively diminishes from the channel surface to the axis, causing a radial rotating flow to form.

Figure 8 depicts the temperature of molten steel on section A-A when the molten steel input temperature is 1750 K. Moreover, in the channel, the temperature of the molten steel increases by 13.90 K, rising from 1752.00 K to 1765.90 K. 

### 3.3. Effect of Molten Steel Temperature on Inclusion Distribution

As determined through studies and practical usage, the receiving chamber eliminates the majority of the inclusions in the double-channel induction-heating tundish. Nonetheless, inclusion particles are still present in the channel and the discharging chamber, significantly affecting the bloom’s quality. Only the movement trajectory of the inclusion particles in the channel and discharging chamber was studied in order to analyze the distribution of inclusion particles in the channel and discharging chamber in detail. Figure 9, Figure 10 and Figure 11 show the computed motion trajectories and dispersion of inclusion particles, respectively, at 11 s, 40 s, 80 s, and 2000 s after the inclusion particles were released at temperatures of 1750 K, 1800 K, and 1850 K. In Figure 9, Figure 10 and Figure 11, the red balls represent inclusion particles, and the blue curves represent the movement trajectories of inclusion particles.

Figure 9, Figure 10 and Figure 11 show how the inclusion particles in the channel travel to the bifurcation and are initially discharged via the bifurcation hole for 11 s. At 40 s, the inclusion particles in region 1 flow out from the bifurcation and impact the inner wall. The inclusion particles in region 2 flow out of the channel and into the middle of the discharge chamber with the molten steel. After the inclusion particles in region 1 impact the inner wall, the inclusion particles continue to flow upward along the molten steel surface, approach the side wall at 80 s, and then move downward along the side wall. Region 2’s inclusion particles are more numerous than those in region 1 at 40 s. When the inclusion particle distribution was compared at 80 s under different molten steel temperatures, it was discovered that the greater the molten steel temperature, the faster the inclusion particles migrate and the more scattered the inclusion particle dispersion. This is because as the temperature of the molten steel rises, its viscosity reduces, and the flow speed of the molten steel rises correspondingly, which facilitates the full flow of inclusion particles in the pouring area. At the same time, the increase in the temperature of the molten steel will reduce its density, thereby reducing the density difference between the density of inclusion particles and the molten steel, which is not conducive to the floating of inclusion particles. However, changes in the density of molten steel mainly affect the flow of molten steel under natural convection conditions, and changes in the viscosity of molten steel have a major impact on the flow of molten steel under turbulent flow conditions. When the temperature of the molten steel was 1750 K, 1800 K, and 1850 K, the average flow velocities of the molten steel in the channel and discharge chamber were 0.0156 m/s, 0.0166 m/s, and 0.0177 m/s, respectively. Therefore, the reduction in molten steel viscosity has a more obvious effect on the flow of molten steel, which is more conducive to the full flow of inclusion particles in the discharging chamber. As a result, the temperature of the molten steel increases, and the removal rate of inclusion particles in the discharge chamber also increases. 

Correspondingly, the distribution of inclusion particles at the same time at higher temperatures is more dispersed. The inclusion particles in the tundish are completely stable at 2000 s. By comparing the inclusion particle distributions under various molten steel temperatures at this time, it was discovered that at 1850 K, there were significantly more inclusion particles in region 2 than at 1750 K and 1800 K, and the movement trajectories were more complex. 

Further analysis revealed that under different molten steel temperature conditions, the inclusions in the channel will form a spiral distribution trend due to the Lorentz force, as seen in Figure 12, in which the blue curve indicates the trend line of the rotational distribution of inclusion particles, and the red dots represent inclusion particles. Since inclusion particles are insulators, they travel toward the channel surface and partially cling to the inner wall when the pinch effect occurs in the channel. This creates a distribution that matches the molten steel’s flow trend. 

Figure 13 compares the inclusion particle removal rates on the channel’s inner surface, the discharging chamber wall surface, the top layer in the discharging chamber, and the outlet under different molten steel temperature settings. The removal rate of inclusion particles on the wall surface of the discharging chamber and the top surface in the discharging chamber increases by 5.40% when the temperature of the molten steel rises from 1750 K to 1850 K. When the temperature of the molten steel rises from 1750 K to 1850 K, the removal percentage of inclusion particles on the discharge chamber’s wall and top layer rises by 9.20%, and the removal percentage in the outlet decreases from 8.00% to 3.00%. On the contrary, when the molten steel temperature is 1850 K, the removal percentage of inclusions on the channel is 4.00% lower than that at 1750 K. In summary, an increase in the molten steel’s temperature will reduce the inclusion removal percentage in the channel, increase the removal percentage in the discharging chamber, and decrease the likelihood that inclusion particles will escape the outlet with the molten steel. As a result, the increase in the temperature of the molten steel is beneficial to the removal of inclusion particles in the discharging chamber and reduces the possibility of inclusion particles flowing out from the tundish outlet.

Above all, this paper mainly studies the influence of different molten steel inlet temperatures on the distribution of inclusion particles, mainly with respect to three temperatures: 1750 K, 1800 K, and 1850 K. The simulation calculation results show that the increase in the temperature of the molten steel will reduce the number of inclusion particles attached in the channel and decrease the number of particles flowing out of the outlet.

## 4. Conclusions

Under electromagnetic-induction-heating conditions, the molten steel in the tundish produced a pinch effect. The magnetic field in the channel rotates with a maximum magnetic flux density of 0.158 T; meanwhile, the Lorentz force is directed toward the center of the axis, with a maximum value of 2.11 × 10^5^ N/m^3^.The increase in the temperature of the molten steel prompted the inclusion particles to move faster. When the temperature of the molten steel increased from 1750 K to 1850 K, the average flow velocity of the molten steel in the channel and discharging chamber accelerated from 0.0156 m/s to 0.0177 m/s. Moreover, the distribution of inclusion particles in the channel was similar to the trend of the rotational flow of the molten steel in the channel.When the molten steel temperature increased from 1750 K to 1850 K, the removal percentage of inclusion particles on the discharging chamber wall and top layer increased by 9.20%, the removal percentage of inclusion particles on the outlet decreased from 8.00% to 3.00%, and the removal percentage on the channel surface decreased from 48.40% to 44.40%.

## Figures and Tables

**Figure 1 materials-16-07556-f001:**
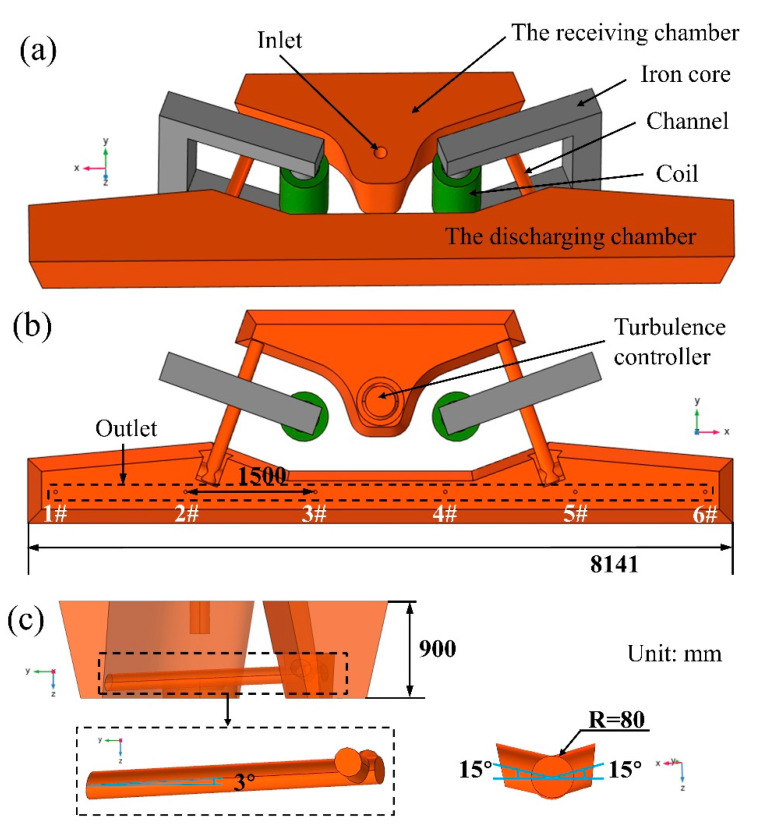
Geometric model of the induction-heating tundish: (**a**) top view; (**b**) bottom view; (**c**) left side view and partial illustration.

**Figure 2 materials-16-07556-f002:**
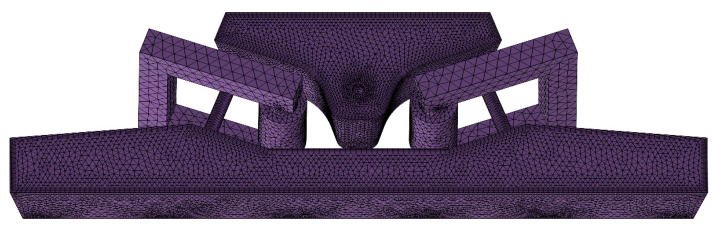
Schematic representation of geometric meshing.

**Figure 3 materials-16-07556-f003:**
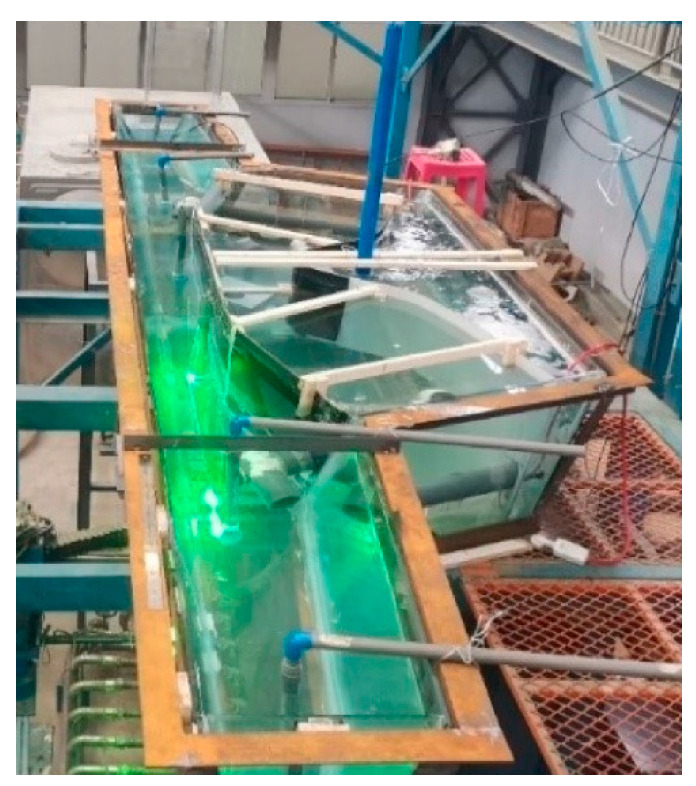
On-site pictures of the six-strand tundish water model.

**Figure 4 materials-16-07556-f004:**
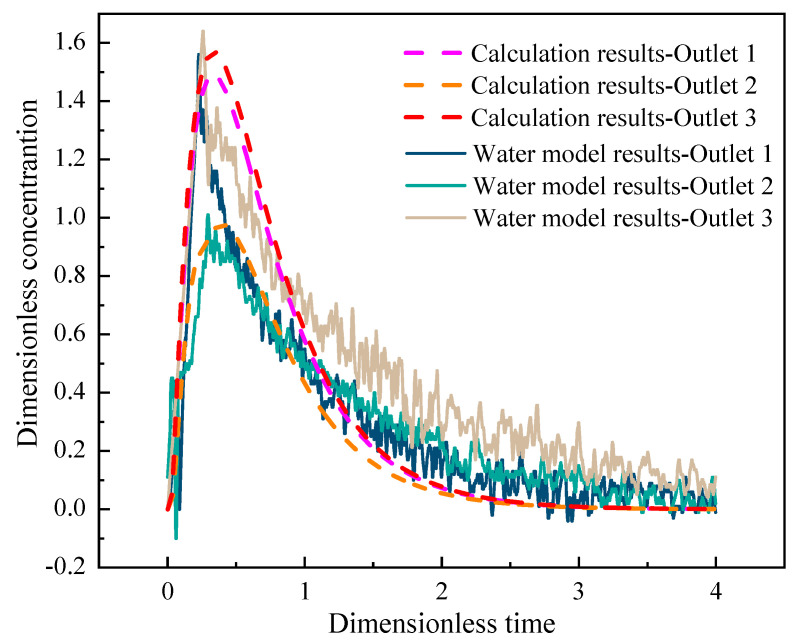
The RTD curve of six-strand induction-heating tundish.

**Figure 5 materials-16-07556-f005:**
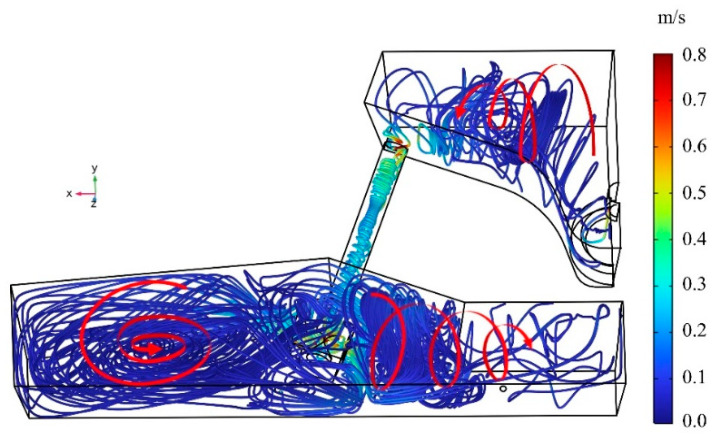
Streamline placement of molten steel in the tundish.

**Figure 6 materials-16-07556-f006:**
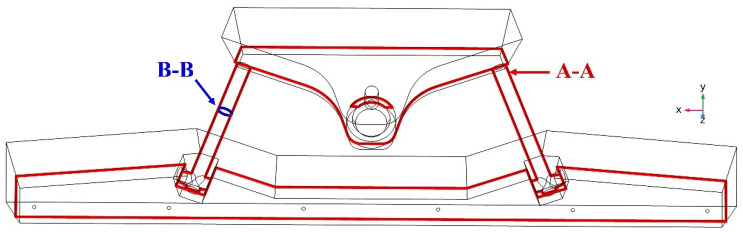
Schematic diagram of tundish cross-section.

**Figure 7 materials-16-07556-f007:**
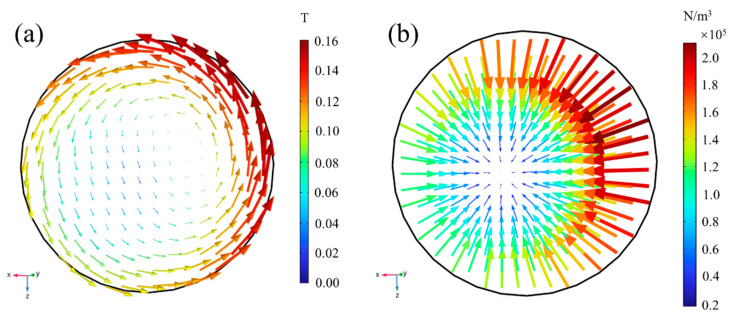
Magnetic flux density distribution (**a**) and Lorentz force distribution (**b**) of molten steel on section B-B.

**Figure 8 materials-16-07556-f008:**
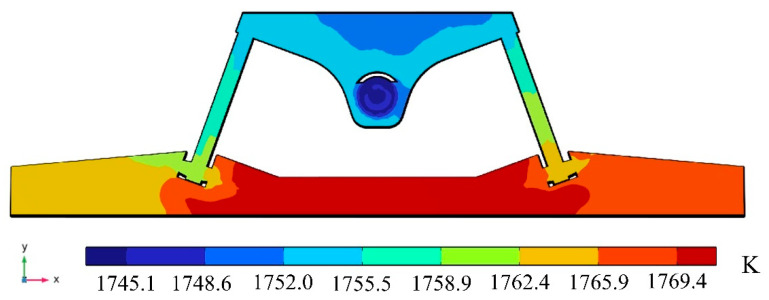
Temperature distribution of molten steel on section A-A.

**Figure 9 materials-16-07556-f009:**
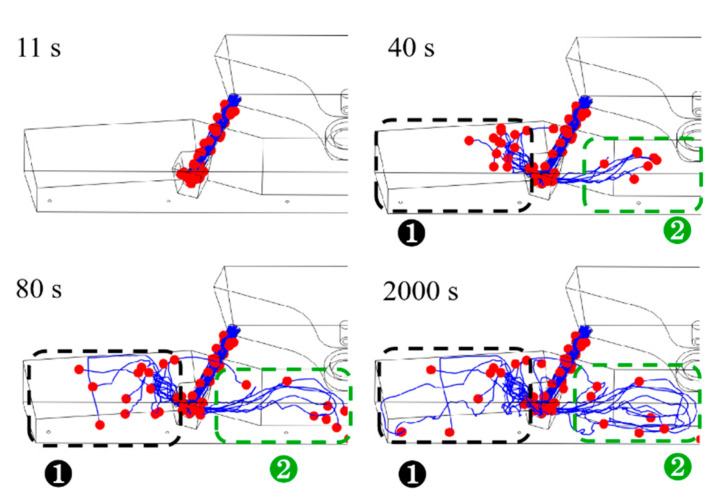
The paths of inclusion particles in the tundish at various points while the molten steel temperature was 1750 K.

**Figure 10 materials-16-07556-f010:**
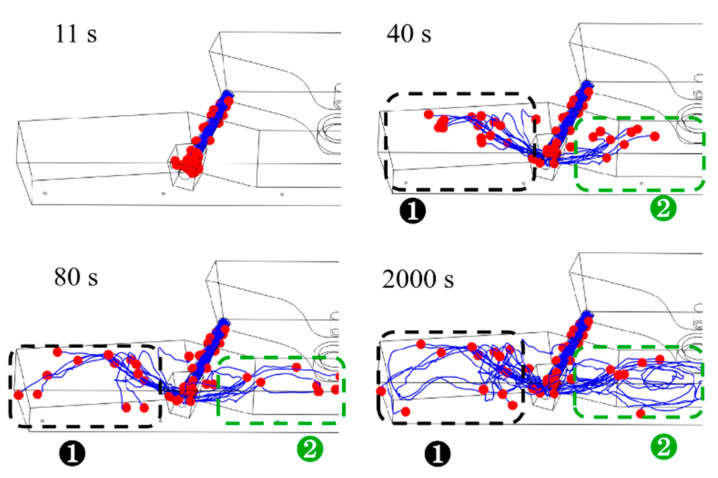
The paths of inclusion particles in the tundish at various points while the molten steel temperature was 1800 K.

**Figure 11 materials-16-07556-f011:**
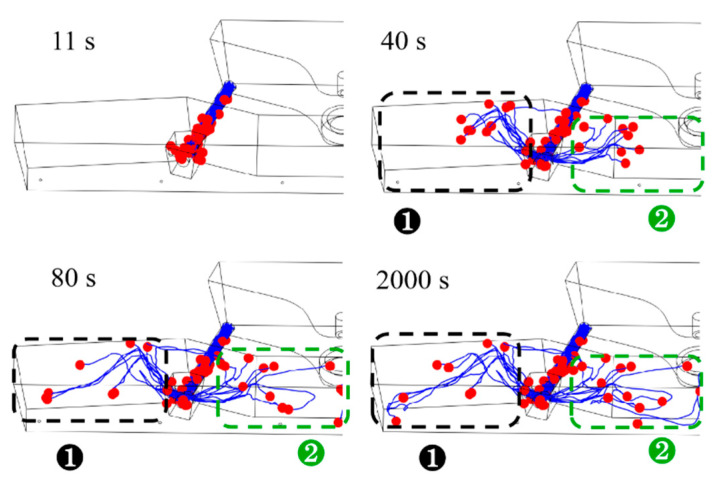
The paths of inclusion particles in the tundish at various points while the molten steel temperature was 1850 K.

**Figure 12 materials-16-07556-f012:**
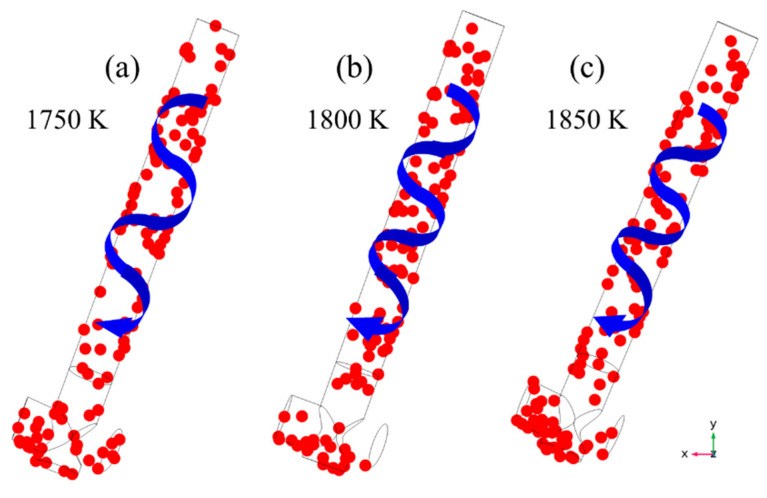
Distribution of inclusion particles in the channel at (**a**) 1750 K, (**b**) 1800 K, and (**c**) 1850 K.

**Figure 13 materials-16-07556-f013:**
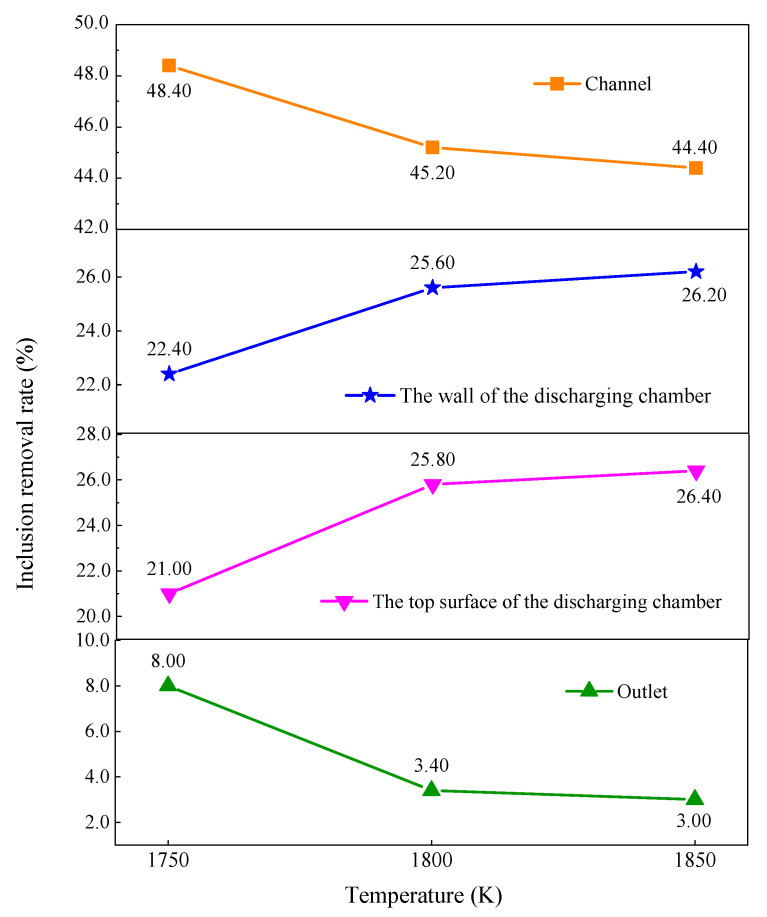
Removal percentages of inclusion particles in different sections of the tundish at various molten steel temperatures.

**Table 1 materials-16-07556-t001:** Detailed geometric dimensions of induction-heating tundish.

Parameter	Value
Radius of inlet (mm)	45
Working level (mm)	900
Length of bottom of receiving chamber (mm)	2863
Length of top of discharging chamber (mm)	8141
Length of main channel (mm)	1630
Radii of main channel and sub-channel (mm)	80
The primary channel’s increase angle (°)	3
The sub-channel’s increase angle (°)	15
The height of the main channel (mm)	145
Radius of outlet (mm)	20
The distance between each outlet (mm)	1500
Number of outlets	6

**Table 2 materials-16-07556-t002:** Detailed geometric dimensions of induction-heating tundish.

	Average Calculation Results			Water Model Results		
	Outlet 1	Outlet 2	Outlet 3	Outlet 1	Outlet 2	Outlet 3
Theoretical residence time (s)	841	864	845	834	901	854
Minimum residence time (s)	20	24	18	18	19	12
Peak time (s)	284	340	286	192	248	218
Dead space volume fraction (%)	1.06	−1.65	0.59	3.36	−4.40	1.04
Piston area volume fraction (%)	17.88	21.41	17.88	12.17	15.47	13.33
Mixing zone volume fraction (%)	81.06	80.24	81.53	84.47	88.93	85.63

## Data Availability

Data are contained within the article.

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
