# Peer review of "Simulation Study on the Influence of Different Molten Steel Temperatures on Inclusion Distribution under Dual-Channel Induction-Heating Conditions"

_materials, 2023, doi:10.3390/ma16247556_

Round 1
Reviewer 1 Report
Comments and Suggestions for Authors
Dear Authors your work require some additional comments.
- Page 2, casting speed is not realistic. I think correct value is 0.68 m/min.
- Table 1, contain mistake in the name of distance between outlets. The word "water: should be delete.
- In page 6, after inlet velocity, initial values for turbulence should be add.
- Page 4, the best mesh should include 1600000 elements not 160000. The word "elements" will be better than "meshes". It will more plainly for Readers.
- "Molten steel" will be more clearly than "melted steel".
- "On site smelting" should be change to "on-site casting" (page 4).
- Staffman force should be change to Saffman force (page 4 and 5).
- Figure 3 should be improve, is not readable in this form.
- The value of heat capacity for liquid steel should be verified.
- Criteria for thermal similarity between numerical model and physical water model should be describe in the work.
- It will be better write "six-strand tundish" not "six-stream tundish".
- Numbering of tundish outlets should be add to figure 1.
- Curves from the section 3.1.1 should be presented in the one figure directly comparing numerical and physical results. Additionally values from both paths i.e. minimum residence time or peak time should be show in additional table. "On-site detection results" is not clear, will be easier write, "water model results".
- At page 10, authors wrote on relation between temperature and viscosity of steel, but presented numerical model not include relation between temperature and viscosity or density. Both values are constant. Therefore influence of temperature on the steel behavior is not considered in the numerical model. This part of work should be more explained.
- The present results and changes due to temperature are very small. Divergence at 1% level are very close to numerical errors, therefore in principle influence of temperature are negligible in this stage. It must be step by step explained in the work.
- What means "inclusion distribution rate" in the figure 13. Please explain in the work.
- Authors should add results on liquid steel velocity for considered cases of inlet liquid steel temperature. This is essential, referred to conclusion no. 2.
Reviewer 2 Report
Comments and Suggestions for Authors
In their manuscript, the authors describe the issue of removing inclusions from molten steel when casting from an induction heating tundish. As they correctly wrote, it is an important process step for improving the quality of the final product, and in this case it is a gate made of continuous casting of molten steel.
The effect of the electromagnetic field on the elimination of impurities is interesting.
Although the research describes continuous steel casting, a water model and CFD simulations are used to obtain the results.
I have a few formal and professional comments on the submitted manuscript:
1) The authors used mathematical relations, which are only a general mathematical description of the processes that were investigated. Where specifically were these equations used if the CFD software COMSOL Multiphysics was used for mathematical modelling? Have these equations been implemented in some way in this software or are they part of it? I ask to explain in the manuscript. I have no comments on the description of the equations themselves.
2) Under the mathematical relation (4), I ask for the correct notation of the dimension of dynamic viscosity, the correct one is "Pa s".
3) The abbreviation "RTD" is used under Figure 3, but without further explanation in the manuscript. I would like a more detailed description of this abbreviation.
4) Figure 4 has an unclear description of the y and x axes. If something is "dimensionless" then it is related to something and such a description is missing here. If time is in "s" how can it be "dimensionless"? I ask to explain in more detail in the text of the manuscript.
I am aware that the authors focused on the effect of different temperatures on the distribution of inclusions, but what effect would a change in the magnetic field have. Apparently, there will be further research in this area. Nevertheless, the manuscript lacks a clearer description of the generated magnetic field. Why is only one magnetic flux value used? Can it be changed in the process or is it set as a constant value?
Round 2
Reviewer 1 Report
Comments and Suggestions for Authors
Dear Authors,
Additional comments are needed.
1. Figure 3 must be reoriented, this point of view is not readable.
2. What means Zb criterion (Text before figure 3).
3. Description of "u" like as flow rate of water inlet is not clear (description of turbulence parameters). U is a inlet velocity. The same for r, it is radius of inlet area.
4. Results in the table 2 for water is not clear. Did You recalculate values from water model by scale factor to full scale steel tundish. The values from water modeling and numerical simulations are very close. Maybe did You simulated by numerical model a water condition?
Reviewer 2 Report
Comments and Suggestions for Authors
The authors revised the manuscript according to the comments and sufficiently answered the ambiguities I mentioned in the review.
After revising the manuscript again, I must state that the authors have substantially improved the manuscript and I have no more serious comments.
Author Response
Thank you for your professional advice to make this article more comprehensive.